# Tropical temperature variability and Kelvin wave activity in the UTLS from GPS RO measurements

Barbara Scherllin-Pirscher[1], William J. Randel[2], and Joowan Kim[2,3]

[1]Wegener Center for Climate and Global Change (WEGC) and Institute for Geophysics, Astrophysics, and Meteorology/Institute of Physics (IGAM/IP), University of Graz, Graz, Austria
[2]National Center for Atmospheric Research (NCAR), Boulder, Colorado, USA
[3]Kongju National University, Gongju, Korea

*Correspondence to:* Barbara Scherllin-Pirscher (barbara.pirscher@uni-graz.at)

**Abstract.** Tropical temperature variability over 10–30 km and associated Kelvin wave activity are investigated using GPS radio occultation (RO) data from January 2002 to December 2014. RO data are a powerful tool to quantify tropical temperature oscillations with short vertical wavelengths due to their high vertical resolution and high accuracy and precision. Gridded temperatures from GPS RO show strongest variability in the tropical tropopause region (on average 3 $K^2$). Large-scale zonal variability is dominated by transient sub-seasonal waves (2 $K^2$), and about half of sub-seasonal variance is explained by eastward traveling Kelvin waves with periods of 4 to 30 days (1 $K^2$). Quasi-stationary waves associated with the annual cycle and inter-annual variability contribute about a third (1 $K^2$) to total resolved zonal variance. Sub-seasonal waves, including Kelvin waves, are highly transient in time. Above 20 km, Kelvin waves are strongly modulated by the quasi-biennial oscillation (QBO) in stratospheric zonal winds, with enhanced wave activity during the westerly shear phase of the QBO. In the tropical tropopause region, however, peaks of Kelvin wave activity are irregularly distributed in time. Several peaks coincide with maxima of zonal variance in tropospheric deep convection, but other episodes are not evidently related. Further investigations of convective forcing and atmospheric background conditions are needed to better understand variability near the tropopause.

## 1 Introduction

Atmospheric temperature variability in the tropics is coupled with dynamical and physical processes, which have crucial impact on the Earth's climate. Variability in the tropical tropopause region is of special interest because it has an important role in troposphere–stratosphere coupling and exchange (Holton et al., 1995; Fueglistaler et al., 2009) and associated global radiation budget. Thus, detailed knowledge of dynamical processes in the tropical upper troposphere and stratosphere is essential for better understanding climate variability and change. It has been found, for example, that decadal variations in global surface climate may be significantly influenced by changes of stratospheric water vapor (Solomon et al., 2010), which is controlled by temperatures near the tropical tropopause (Mote et al., 1996). Furthermore, vertically propagating equatorial waves, which originate from the troposphere and propagate into the upper atmosphere, are main drivers of global-scale zonal wind variation, which is called the quasi-biennial oscillation (QBO). The QBO manifests itself as alternating easterly and westerly zonal winds

in the tropical stratosphere with a period of approximately 28 months. It has large impacts covering the global atmosphere (Baldwin et al., 2001).

Equatorial atmospheric waves, including eastward propagating Kelvin waves, are generated by heating associated with convection (e.g., Garcia and Salby, 1987; Wheeler and Kiladis, 1999).

Kelvin waves in the troposphere have been identified with convection (Wheeler and Kiladis, 1999; Wheeler et al., 2000). These features have zonal wavenumbers ∼3–7 and periods ∼4–10 days, and move coherently with convection. Above the troposphere, Kelvin waves are observed with planetary scales (zonal wavenumbers 1 to 5), periods from a few days to a few tens of days, and vertical wavelengths of a couple of kilometers to more than 10 km (Andrews et al., 1987; Tsuda et al., 1994; Randel and Wu, 2005; Alexander and Ortland, 2010). They are equatorially trapped and perturb the zonal wind and

temperature fields. Atmospheric Kelvin waves were theorized by Matsuno (1966) and Holton and Lindzen (1968) and first observed by Wallace and Kousky (1968). Kelvin waves can propagate vertically in regions of background easterly winds, while they become trapped below regions of westerly winds, and they exhibit strong interactions with the stratospheric QBO (e.g., Sato and Dunkerton, 1997).

Kelvin waves dominate sub-seasonal variability in the tropical tropopause region (Tindall et al., 2006; Kim and Son, 2012).

They modulate the height and temperature of the tropical tropopause (Tsuda et al., 1994; Shimizu and Tsuda, 1997) and were found to be important for cirrus formation (Boehm and Verlinde, 2000; Immler et al., 2008; Suzuki et al., 2013) and dehydration of air entering the lower stratosphere (Fujiwara et al., 2001). Furthermore, they play an important role in stratosphere–troposphere exchange of ozone (Fujiwara et al., 1998).

Kelvin waves have been identified using radiosonde wind and temperature measurements (e.g., Wallace and Kousky, 1968;

Tsuda et al., 1994; Holton et al., 2001). Due to the limited spatial coverage of the radiosonde network, observations were confined to a few specific locations. Later, satellite temperature measurements provided a more global view on atmospheric dynamics. Equatorial atmospheric waves have been investigated using satellite data from Nimbus-7 LIMS (Limb Infrared Monitor of the Stratosphere) (Salby et al., 1984), the Microwave Limb Sounder (MLS) (Mote et al., 2002), SABER (Sounding of the Atmosphere using Broadband Emission Radiometry) (Ern et al., 2008; Ern and Preusse, 2009), and High Resolution

Dynamics Limb Sounder (HIRDLS) (Alexander and Ortland, 2010). Furthermore, several studies on atmospheric Kelvin waves are based on analysis or reanalysis data from numerical weather prediction (NWP) centers (Tindall et al., 2006; Suzuki and Shiotani, 2008; Suzuki et al., 2010; Fujiwara et al., 2012; Flannaghan and Fueglistaler, 2013).

Since 2001, highly accurate temperature soundings have continuously been available from Global Positioning System (GPS) radio occultation (RO) measurements. Due to their characteristics of high vertical resolution, accuracy, and global coverage

(see Sect. 2.1 for more details) these data have been extensively used to analyze temperature variability and Kelvin wave activity in the upper troposphere and lower stratosphere (UTLS) region (Tsai et al., 2004; Randel and Wu, 2005; Ratnam et al., 2006; Gettelman and Birner, 2007; Alexander et al., 2008; Steiner et al., 2011; Kim and Son, 2012; Scherllin-Pirscher et al., 2012; Das and Pan, 2013; Flannaghan and Fueglistaler, 2013; Randel and Wu, 2015).

In this study we utilize the entire RO record from 2002 to 2014 to investigate long-term behavior of zonal temperature

variability and Kelvin wave activity over altitudes 10–30 km. We use GPS RO data from Challenging Mini-Satellite Payload

(CHAMP) (Wickert et al., 2001), Gravity Recovery and Climate Experiment (GRACE) (Wickert et al., 2005), Satélite de Aplicaciones Científicas (SAC-C) (Hajj et al., 2004), and Formosa Satellite mission-3/Constellation Observing System for Meteorology, Ionosphere, and Climate (Formosat-3/COSMIC) (Anthes et al., 2008).

## 2 Data and method

### 2.1 Radio occultation data

We utilize atmospheric profiles from GPS RO observations to characterize tropical temperature variability in the UTLS region. GPS RO temperature retrievals are characterized by high vertical resolution (from about 100 m in the troposphere (Gorbunov et al., 2004) to about 1.5 km in the stratosphere (Kursinski et al., 1997)) and high accuracy (0.7 K to 1 K between 8 km and 25 km (Scherllin-Pirscher et al., 2011b)). Since measurements can be obtained during day and night as well as in nearly any meteorological weather conditions, data are available with good coverage in space and time (Anthes, 2011).

RO data used in this study were processed at the Wegener Center for Climate and Global Change (WEGC) using the Occultation Processing System (OPS) version 5.6 (Schwärz et al., 2013). The record consists of 13 years of data extending from January 2002 to December 2014. It includes measurements from CHAMP (2002 to 2008), GRACE (2007 to 2014), SAC-C (2006 to 2011), and Formosat-3/COSMIC (2006 to 2014). Due to the RO measurement principle these data from different satellites can be merged to a single observational record without the need for explicit calibration or homogenization (Hajj et al., 2004; Schreiner et al., 2007; Foelsche et al., 2011).

However, the number of available RO profiles is not constant over time, because from 2002 to 2006 there were only measurements available from CHAMP. During this time period, there were only about 400 RO profiles available between $10°S$ and $10°N$ per month. This number significantly increased after 2006, when more than 5000 tropical RO profiles per month can be exploited (see Fig. 1 of Scherllin-Pirscher et al., 2011a).

We utilize atmospheric profiles of dry temperature, which is the same as physical temperature if humidity is negligible. Since we investigate temperature variability only above 10 km where humidity effects are small, dry temperature can be used as proxy for temperature (Scherllin-Pirscher et al., 2011a).

### 2.2 Additional data sets

We use information on convection and background winds to evaluate relationships with temperature variability observed in GPS RO data. As a proxy for convective activity we use daily gridded fields of outgoing long-wave radiation (OLR) on a $2.5° \times 2.5°$ latitude-longitude grid provided by the National Oceanic and Atmospheric Administration (NOAA) (Liebmann and Smith, 1996). For quantifying background zonal winds in the tropics we use vertically-resolved zonal winds above Singapore, which provides a standard index for the QBO. These monthly mean wind profiles are provided by the Freie Universität Berlin (FUB) and are available on standard pressure levels spanning 100–10 hPa. We interpolate these data to a vertical altitude grid assuming a scale height of 7 km and a surface pressure of 1013 hPa.

## 2.3 Gridding and spectral analysis

Tropical (10°S to 10°N) temperature profiles from GPS RO are gridded in longitude, altitude, and time (no further latitudinal gridding). Following Randel and Wu (2005), we calculate daily mean fields with a longitudinal resolution of $\Delta\lambda = 30°$ and a vertical resolution of $\Delta z = 100$ m. Since underlying GPS RO profiles have a vertical resolution of 0.1–1.5 km (see Sect. 2.1), adjacent vertical levels are not fully independent of each other. Due to the relatively small number of RO measurements before 2006 (when only CHAMP measurements are used), we also include data from two neighboring days ($\pm 2$ days) and apply a weighted temporal average (Gaussian weighted with a 1-day e-folding time). On average, there are approximately 6 profiles per grid box per day during the CHAMP-only period. This number significantly increased to more than 55 profiles per grid box per day after 2006. Infrequent missing grid points are interpolated using profiles from neighboring (longitude × time) grid cells. These gridded data can resolve waves with zonal wavenumbers up to 6. While this sampling strategy accurately resolves traveling planetary waves with periods longer than 10 days, amplitudes of waves with short periods ($<5$ days) are underestimated or poorly resolved (see Randel and Wu, 2005, for details).

These sampling details are important because they will affect all subsequent calculations. Testing different spatio-temporal resolutions reveals that changing the temporal resolution of the grid cells to only $\pm 1$ day results in too many empty grid boxes during the CHAMP-only period. However, these tests also reveal that the large-scale temporal evolution is essentially the same before and after 2006, independent of including data from $\pm 1$ day or $\pm 2$ days.

To quantify behavior of atmospheric waves we apply space-time spectral analysis (Hayashi, 1971, 1982) separately at each height level using the entire RO record. The record from January 2002 to December 2014 has total length of 4748 days. Figure 1 shows the wavenumber–frequency spectrum of temperature at an altitude of 18 km, based on smoothing the raw spectra in the frequency domain with a Gaussian filter (with an e-folding width of ten frequency bins). The spectrum reveals a maximum for low frequencies and wavenumber zero, which is largely due to the zonally-symmetric annual cycle, driven by the Brewer-Dobson circulation (BDC), and the QBO. The maximum zonal wave power at 18 km occurs for eastward propagating waves with wavenumbers one and two at periods between $\sim$8–30 days. These waves are eastward propagating equatorial Kelvin waves. A smaller signal related to the Madden-Julian oscillation (MJO) occurs for wavenumbers 0 to +4 at approximately 50 days (Madden and Julian, 1971).

We furthermore decompose temperature variability into low-frequency and sub-seasonal components, with filtering based on direct Fourier transforms. Low-frequency variations are defined as having periods larger than 100 days. They include contributions of slow and quasi-stationary variations such as the annual cycle and inter-annual variability (El Niño–Southern Oscillation (ENSO) and the QBO). Sub-seasonal variability (periods shorter than 100 days) essentially contains signals of the MJO and equatorial waves (e.g., Wheeler and Kiladis, 1999; Ern et al., 2008). We isolate Kelvin wave activity by selecting wavenumbers $k = +1$ to $+6$, periods from 4 to 30 days, and equivalent depths from 8 m to 240 m. After filtering, reverse Fast Fourier Transformations (FFTs) are applied to reconstruct atmospheric variability associated with individual types of waves. In order to investigate temporal variations of Kelvin wave activity we apply wavelet analysis as described by Torrence and Compo (1998) using the Morlet wavelet basis function.

# 3 Results and discussion

## 3.1 Spatial and temporal characteristics of temperature variability

As an example, zonal structure of tropical temperature anomaly from GPS RO is shown for one day in January 2010 along with the behavior isolated for Kelvin waves (Fig. 2). The Kelvin wave captures much of the structure over 15–30 km, where both patterns show an eastward phase-tilt with height that is a characteristic feature of upward propagating Kelvin waves (e.g., Wheeler et al., 2000; Ryu et al., 2008). Strongest amplitudes are observed near the tropopause, where the magnitude of zonal anomalies (i.e., anomalies relative to the daily-mean zonal-mean) exceeds 5 K and that of Kelvin waves exceeds 2 K. The strong positive temperature anomaly close to 75°E is associated with a depression of the tropopause altitude of almost 2 km. The vertical profile of associated variance in Fig. 2 shows a peak of resolved variance ($>8$ K$^2$) between 17 km and 18 km. Kelvin wave variance shows a maximum at 18 km to 19 km but it is only about a quarter of resolved variance ($>2$ K$^2$), while a large fraction of the variance in the stratosphere is associated with Kelvin waves. Temperature anomalies in the troposphere do not exhibit a phase-tilted structure (Fig. 2a). Rather, the patterns show warm anomalies extending into the upper troposphere (around 14 km) and centered close to the date line. Kelvin wave activity is weak in this region (Fig. 2b).

Figure 2 reveals that Kelvin waves may contribute only a fraction of variability in the tropical tropopause region on an individual daily basis. So, what else contributes to total zonal temperature variability in this region? Figure 3 shows temporal evolution of zonal temperature anomalies (relative to daily-mean zonal-mean) from November 2009 to February 2011 at 18 km. Sub-seasonal and low-frequency components of the anomalies are also shown. Both zonal and sub-seasonal anomalies (Fig. 3a,b) highlight eastward propagating wave events linked to Kelvin waves. Roughly half of the variance in sub-seasonal anomaly is explained by Kelvin waves at this level. Enhanced Kelvin wave activity is found in January, May, and August to October 2010.

Quasi-stationary low-frequency variations are also a large part of total temperature variability near the tropopause (Fig. 3c). These are tied to low-frequency variations in tropical convection (Randel et al., 2003; Gettelman and Birner, 2007) and have annual and inter-annual variations near the tropopause. At 18 km negative temperature anomalies are evident east of the region of enhanced convection. While maximum low-frequency temperature variance is usually found in boreal winter (see below), some differences in anomaly patterns are found between 2009/2010 and 2010/2011 due to ENSO. A moderate El Niño event in late 2009/early 2010 shifted convective regions towards the eastern equatorial Pacific, while a moderate La Niña event in late 2010/early 2011 shifted equatorial convective regions to the western part of the Pacific basin. The negative temperature anomalies at 18 km correspond well with the shift of convective regions caused by different phases of ENSO.

The annual cycle is an important component in low-frequency variations near the tropopause. Temporal and spatial characteristics of the mean annual cycle of low-frequency temperature anomalies (averaged over all years of data) are shown in Fig. 4. At an altitude of 18 km largest temperature anomalies are observed from November to May with strong negative anomalies east of the convective regions of the maritime continent and over the western Pacific (approximately 120°E to 210°E). A second but significantly weaker temperature minimum is observed above South America (close to 285°E).

The zonal cross-sections of low-frequency temperature anomalies in December-January-February (DJF, Fig. 4b) and June-July-August (JJA, Fig. 4c) reveal that maximum temperature anomalies occur near the tropopause (in DJF) or slightly below (in JJA). Positive temperature anomalies in the troposphere coincide with negative anomalies close to the tropopause and vice versa. The transition between warming and cooling occurs near 14 km, roughly at the level of zero radiative heating (Gettelman and Forster, 2002). In DJF, negative temperature anomalies close to the tropopause (which tilt slightly eastward with height) are distinctively larger than in JJA, and this is reflected in the seasonal variation at 18 km seen in Fig. 4a.

These results show that quasi-stationary patterns are primarily responsible for differences between zonal- and Kelvin wave anomalies in the tropical tropopause region as shown in Fig. 2.

## 3.2 Long-term characteristics of temperature variability

Long-term variability in these data is analyzed based on daily vertical profiles of zonal mean variances for the entire RO record from January 2002 to December 2014. Figure 5 shows time series for resolved variance and Kelvin wave variance for the long-term record. To put temperature variance in context of background conditions, the top panel of Fig. 5 shows zonal mean wind speed above Singapore, which highlights downward propagating QBO variations.

In the lower and middle stratosphere (above approximately 20 km), there is a strong modulation of temperature variance by the QBO. Enhanced temperature variance is observed during the westerly shear phase of the QBO (where the QBO winds switch from easterly to westerly with altitude), while this is not evident during the easterly shear phase. Since variance is mainly caused by sub-seasonal fluctuations, enhanced variance is found in both resolved variance and Kelvin wave variance. In fact, Kelvin wave activity dominates the gridded variance at these levels.

A relative maximum in variance is found in the tropical tropopause region between approximately 16 km and 20 km in Fig. 5b. However, temporal evolution of the variance is not associated with the QBO but shows an annual cycle with maxima in boreal winter (see Fig. 4). Kelvin waves contribute to enhanced variance near the tropopause but their amplitudes do not exhibit any distinct periodicities (as shown below). Small Kelvin wave variance is found below the tropopause.

Figure 5b shows distinctively larger temperature variability before 2006 compared to after 2006. This behavior is related to the smaller number of RO measurements before 2006 (during the CHAMP-only period). A similar jump is less evident in Kelvin wave variance. We have tested the effect of GPS RO sampling on gridded temperature variance, based on sub-sampling the period with dense observations (after 2006) using just one satellite (Formosat-3/COSMIC flight model 1 (FM-1) from July 2006 to December 2014). Comparisons of variances obtained from all RO measurements with the sub-sampled set of RO measurements (Fig. 6) show that resolved variance and sub-seasonal variance estimates are sensitive to the number of data included in the samples, with higher variances for sampling by only one GPS RO satellite. This can be explained by noting that much of sub-seasonal variance is related to sub-grid-scale (smaller than $30°$) variation (as shown in Randel and Wu, 2005), which is under-sampled by one satellite. This means that the relatively large variance before 2006 as shown in Fig. 5b is due to the lack of dense measurements. Due to this change in sampling, it is not possible to combine daily variance from a single satellite with daily variance from multiple satellites to quantify long-term changes.

In contrast, comparisons in Fig. 6 for low-frequency variance and Kelvin waves show relatively small differences for sampling between one and several GPS RO satellites. This is because even one satellite can accurately resolve low-frequency planetary-scale features. These results suggest that it is possible to combine RO data and use the full record from 2002 onward to analyze Kelvin wave activity.

Comparison of mean vertical profiles of resolved variance, low-frequency and sub-seasonal variance as well as Kelvin wave variance obtained from all satellite measurements (thick profiles in Fig. 6) shows the relative importance of individual components. In the lower and middle stratosphere (above $\sim$19 km) low-frequency variance is negligible, and variance is mainly caused by sub-seasonal fluctuations. Kelvin waves dominate these sub-seasonal fluctuations, and they contribute approximately 65 % of the resolved variance in the lower and middle stratosphere. Larger differences between resolved variance and sub-seasonal variance are observed below 19 km. Even though both types of variance peak somewhere between 17.5 km and 18 km, sub-seasonal variance is significantly smaller than resolved variance with means of 2 $K^2$ and 3 $K^2$, respectively. Low-frequency variance also increases below 19 km and reaches a maximum of about 1.2 $K^2$ at 17 km. This is due to the influence of quasi-stationary waves in boreal winter (see Figs. 3 and 4).

Kelvin wave activity peaks near 18 km, where its mean variance amounts to approximately 1.2 $K^2$. This is about half of sub-seasonal variance. Ryu et al. (2008) suggest that the height of maximum Kelvin wave activity slightly above the tropical tropopause is due to the rapid increase of static stability above the tropopause and only weak dependence on background wind speed in this region. Kelvin wave activity decreases below 18 km; near the tropopause at 17 km it is only 50 % of sub-seasonal variance, and less than 40 % below.

## 3.3   Temporal variations in Kelvin wave activity

In the lower and middle stratosphere Kelvin wave activity is maximum during the westerly shear phases of the QBO (see Fig. 5c). This behavior is highlighted in Fig. 7, showing temporal variations of daily Kelvin wave variance at 25 km. The smoothed time series is obtained by applying a 61-day moving average. Kelvin wave variance in Fig. 7a shows amplitude variations that are strongly modulated by the QBO, with enhanced wave activity in periods of transition from easterly to westerly stratospheric wind (westerly shear zones). Similar results have been found by Randel and Wu (2005); Ern et al. (2008); Alexander and Ortland (2010); Flannaghan and Fueglistaler (2013). These peaks in Kelvin wave activity are expected since Kelvin wave energy is accumulated below the critical level, which is located near the zero-wind line. The wavelet power spectrum (Fig. 7b) confirms that most power is concentrated at periods from 24 to 32 months, which corresponds to the period of the QBO. Some wavelet power at periods between 12 and 16 months is associated with some smaller peaks of wave activity, in particular at the end of the time series.

While Kelvin wave activity is clearly associated with the QBO above 20 km, variability in the vicinity of the tropical tropopause shows less regularity. Figure 8 shows detailed time variations of daily Kelvin wave variance between 16 km and 20 km. Weak Kelvin wave activity is observed at 16 km. The wavelet power spectrum (not shown) reveals enhanced power at a period of approximately 12 months from 2003 to 2007 and from 2010 to 2012, which is in agreement with Alexander and

Ortland (2010), who attributed this annual variation to effects of the background wind and stability on Kelvin wave propagation in the tropical tropopause layer.

Kelvin wave activity reaches maximum amplitude around 18–19 km (see Fig. 6), and variability between 18 km and 20 km is highly correlated across these levels. The peak in early 2004, which is the largest in the GPS RO record, is evident at each level. In general, peaks of enhanced Kelvin wave activity are irregularly distributed in time.

To assess whether this temporal variability should be attributed to temporal variations of the tropopause rather than to Kelvin wave activity itself, we calculated Kelvin wave variance in cold-point tropopause coordinates. Figure 9 shows results at the cold-point tropopause and two kilometers above. Comparison to Fig. 8 shows very similar temporal evolutions. The time series at the tropopause and 2 km above (Fig. 9) are virtually similar to those at 17 km and 19 km in altitude coordinate (Fig. 8), respectively. Again, no clear periodicity of Kelvin wave activity can be found.

Figure 10 shows the wavelet power spectrum for the Kelvin wave amplitudes at 19 km. There is a peak in wavelet power in boreal spring 2004 at a period of approximately one year, linked to the maximum wave activity in early 2004, together with two smaller peaks in boreal spring 2003 and 2005 (see Figs. 8 and 11). Maximum wave activity in 2004 could be related to the QBO westerly shear phase (see Fig. 5), although other periods of westerly shear near the tropopause do not show such large wave amplitudes.

Enhanced wavelet power is also observed from 2009 to 2013. It has a period of approximately 9 months, which slightly shifts towards longer periods (about one year) at the end of the time series. The approximately 9-month period is caused by enhanced Kelvin wave activity in April 2009, January 2010, October 2010, June 2011, June 2012, and February 2013 as observed in Fig. 8. This 9-month periodicity is not observed from 2002 to 2008. The use of a shorter observational record (such as the Formosat-3/COSMIC record from 2006 onward) could therefore lead to a misleading interpretation of the month-to-month variability of Kelvin wave activity.

What causes the month-to-month variability of Kelvin wave activity near the tropopause? Theoretical and modeling studies and previous observational studies (cited above) suggest Kelvin waves should be influenced by convective forcing and changes in background winds and stability, and hence we investigated these quantities to explain the variations seen in Fig. 8 and 9. An example of the relationship with convective forcing is shown in Fig. 11, showing time series of Kelvin wave variance at 19 km and time series of zonal variances of high-pass filtered OLR data between 10°S and 10°N. These high-pass filtered OLR data are obtained by applying a 100-day Fourier filter at each grid point. Large zonal variances indicate enhanced variability from short-term fluctuations and different types of waves similar to high-pass filtered temperature anomalies shown in Fig. 3b.

Zonal variance of filtered OLR data (Fig. 11b) has a pronounced annual cycle, which peaks in northern hemisphere spring (April/May). Almost every year, there is a second peak in fall or early winter (November/December). Several peaks in Kelvin wave activity match peaks in OLR variance. However, there are also several mismatches, where OLR variability is large but Kelvin wave activity is weak (spring 2003, 2005, 2007, and 2014) and also several mismatches were OLR variability is small but Kelvin wave activity is large (fall 2006, fall 2010). Another discrepancy between Kelvin wave activity and OLR variability is that the former has strong month-to-month variability while the latter peaks have similar amplitude every year. Discrepancies between equatorial wave activity close to the tropopause and wave activity in tropospheric convection have

also been found by Alexander et al. (2008) and Alexander and Ortland (2010). Suzuki and Shiotani (2008) suggested that the background zonal wind field modulates the propagation of these waves. This was also found by Ryu et al. (2008), who showed that background zonal wind plays an important role in modulating Kelvin waves close to the tropical tropopause. More recently, Flannaghan and Fueglistaler (2013) showed that seasonal and inter-annual variability of Kelvin wave propagation is dominated by the variability in the background wind field. However, we have explored this behavior and do not find any evident relationships between Kelvin wave amplitudes and changes in winds near the tropopause (based on Singapore winds). More detailed calculations may need to follow Flannaghan and Fueglistaler (2013) and use the full background structure along the waves' trajectories to determine their amplitudes.

## 4 Summary and conclusions

Using 13 years of GPS radio occultation (RO) data we have investigated tropical temperature variability and associated Kelvin wave activity in the upper troposphere and lower stratosphere (UTLS) region. In this region, RO measurements are characterized by high accuracy and precision as well as high vertical resolution, which makes these data ideal for characterizing temperature oscillations with short vertical wavelengths.

We have constructed daily gridded temperature fields in the tropics (10°N to 10°S) from January 2002 to December 2014, and examined variability on fast and slow time scales (periods shorter and longer than 100 days). Eastward traveling Kelvin waves are an obvious feature in these data (e.g. Fig. 1), and we use space-time spectral analysis to isolate Kelvin waves with zonal waves +1 to +6, periods of 4 to 30 days, and equivalent depths of 8 m to 240 m.

Largest zonal temperature variability ("resolved variance") was found in the tropical tropopause region close to the tropopause. Maximum variance (3 $K^2$) was found between 17.5 km and 18 km. Quasi-stationary waves with periods larger than 100 days are an important part of zonal variability in this region, and there is a strong annual cycle with maximum amplitude during boreal winter (Fig. 4). Low-frequency inter-annual variability is also associated with the El Niño–Southern Oscillation (ENSO). These low-frequency waves are strongly tied to convection (Figs. 3 and 4). Tropospheric temperature is higher eastward of regions of strong convective activity (e.g., above the maritime continent and the western Pacific). ENSO activity slightly shifts these centers of convection and temperature response. Transition from warming to cooling occurs close to 14 km and distinct negative anomalies are observed east of convective region close to the tropopause. Low-frequency wave activity maximizes near the tropical tropopause (1.2 $K^2$ at approximately 17 km).

Sub-seasonal waves (periods <100 days) dominate zonal temperature variability above the tropical tropopause. Maximum wave activity (2 $K^2$) was found slightly below 18 km. In the lower and middle stratosphere (above ~20 km) this temperature variance is strongly modulated by the QBO, with enhanced wave activity observed during the westerly shear phase of the QBO (Figs. 5 and 7). Transient Kelvin waves are an important part of sub-seasonal waves. They contribute approximately 65 % of the resolved variance above 20 km. Maximum Kelvin wave activity (1.2 $K^2$) was found at 18 km, decreasing at lower altitudes (to less than 0.4 $K^2$ at 16 km).

Another aspect of this study was to investigate long-term (13 years) characteristics of tropical temperature variability. However, the number of available RO measurements is not constant with time, but increased significantly in 2006 after the launch of the multi-satellite mission Formosat-3/COSMIC. We quantified the influence of changes in the number of RO measurements and found increased variance in gridded data due to the lack of dense measurements before 2006 (Fig. 6). Therefore, it is not possible to combine daily variances from a single satellite with daily variances from multiple satellites. However, there are relatively small differences for analysis of low-frequency or planetary-scale Kelvin waves, as these are sampled well by even one GPS RO satellite. Hence we are confident to use the entire 13-year record of RO to investigate Kelvin wave activity.

In general, Kelvin waves show strong amplitude variations over time. Above 20 km, enhanced Kelvin wave activity is found during the westerly shear phase of the QBO. However, near the tropopause (∼16 km to 20 km) peaks of enhanced wave activity are irregularly distributed in time without a distinct periodicity. At 19 km (close to the level where maximum Kelvin wave activity occurs), we found 6 distinct peaks with an approximately 9-month period between 2009 and 2013. This 9-month period, however, was not observed during 2002 and 2008.

We further explored the influence of deep convective activity in the tropical troposphere on Kelvin wave activity. We found that several peaks in Kelvin wave activity coincide with peaks of zonal variance of sub-seasonal waves of convective activity but other maxima are not evidently related. Also, there are no obvious relationships with zonal winds or stability fields near the tropopause level. Hence the nature of the modulations in Kelvin waves near the tropopause remains poorly understood. One important step towards a better understanding could be to follow Flannaghan and Fueglistaler (2013) and use the full background structure along the waves' trajectories.

*Author contributions.* B.S.-P. collected the data, performed the analyses, and wrote the manuscript. W.J.R. and J.K. provided guidance on all aspects of the study and contributed to the text.

*Acknowledgements.* We are grateful to the UCAR/CDAAC and WEGC RO processing team members. Especially M. Schwärz is thanked for OPSv5.6 RO data provision. Furthermore, we want to thank NOAA for providing OLR data and FU Berlin for Singapore zonal winds. We thank F. Ladstädter (WEGC, AT), A. K. Steiner (WEGC, AT), and R. Garcia (NCAR, USA) for helpful comments and input. This work was funded by the Austrian Science Fund (FWF) under research grant T620-N29 (DYNOCC).

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

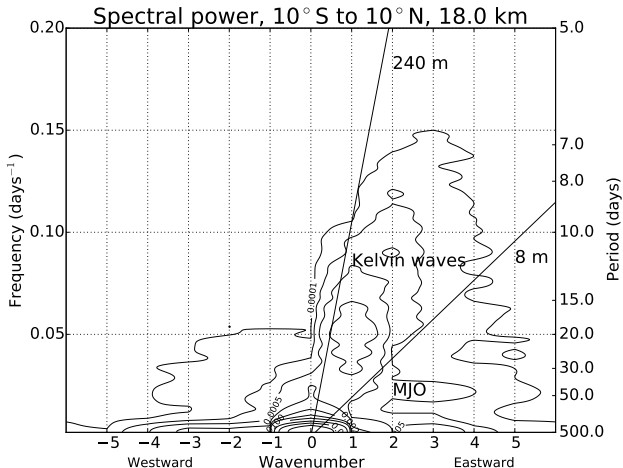

**Figure 1.** Zonal wavenumber–frequency power spectrum of RO temperature at 18 km, calculated for the entire record from January 2002 to December 2014. After spectral analysis, the frequency domain of the spectrum is smoothed with Gaussian filter. Logarithmic contour interval is (0.0001, 0.00025, 0.0005, 0.001,..., 10, 25, 50) K$^2$. The two straight lines indicate equivalent depths of 8 m and 240 m.

Tsai, H.-F., Tsuda, T., Hajj, G. A., Wickert, J., and Aoyama, Y.: Equatorial Kelvin Waves Observed with GPS Occultation Measurements (CHAMP and SAC-C), J. Meteor. Soc. Japan, 82, 397–406, 2004.

Tsuda, T., Murayama, Y., Wiryosumarto, H., Harijono, S. W. B., and Kato, S.: Radiosonde observations of equatorial atmosphere dynamics over Indonesia: 1. Equatorial waves and diurnal tides, J. Geophys. Res., 99, 10 491–10 505, doi:10.1029/94JD00355, 1994.

5  Wallace, J. M. and Kousky, V. E.: Observational Evidence of Kelvin Waves in the Tropical Stratosphere, J. Atmos. Sci., 25, 900–907, 1968.

Wheeler, M. and Kiladis, G. N.: Convectively Coupled Equatorial Waves: Analysis of Clouds and Temperature in the Wavenumber–Frequency Domain, J. Atmos. Sci., 56, 374–399, 1999.

Wheeler, M., Kiladis, G. N., and Webster, P. J.: Large-scale dynamical fields associated with convectively-coupled equatorial waves, J. Atmos. Sci., 57, 613–640, 2000.

10  Wickert, J., Reigber, C., Beyerle, G., König, R., Marquardt, C., Schmidt, T., Grunwaldt, L., Galas, R., Meehan, T., Melbourne, W., and Hocke, K.: Atmosphere sounding by GPS radio occultation: First results from CHAMP, Geophys. Res. Lett., 28, 3263–3266, 2001.

Wickert, J., Beyerle, G., König, R., Heise, S., Grunwaldt, L., Michalak, G., Reigber, C., and Schmidt, T.: GPS radio occultation with CHAMP and GRACE: A first look at a new and promising satellite configuration for global atmospheric sounding, Ann. Geophys., 23, 653–658, 2005.

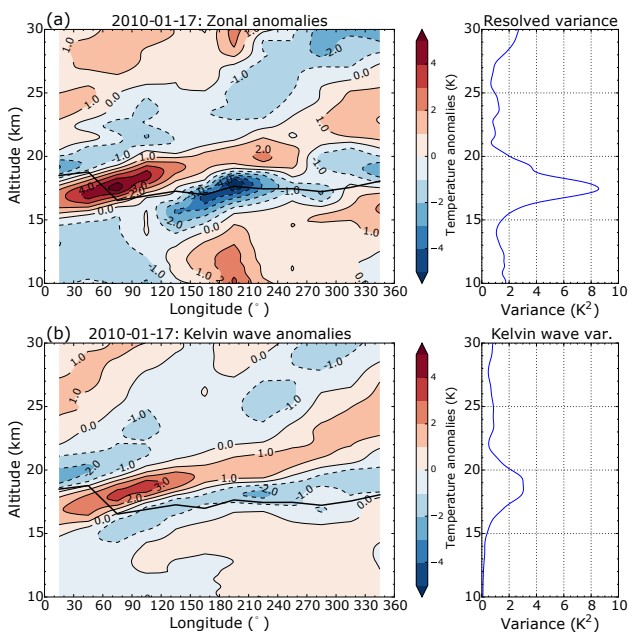

**Figure 2.** Zonal (top) and Kelvin wave (bottom) temperature anomalies as a function of longitude and altitude from January 17, 2010. The black thick line indicates the lapse-rate tropopause. Vertical profiles of associated zonal mean variances are plotted on the right hand side of each panel.

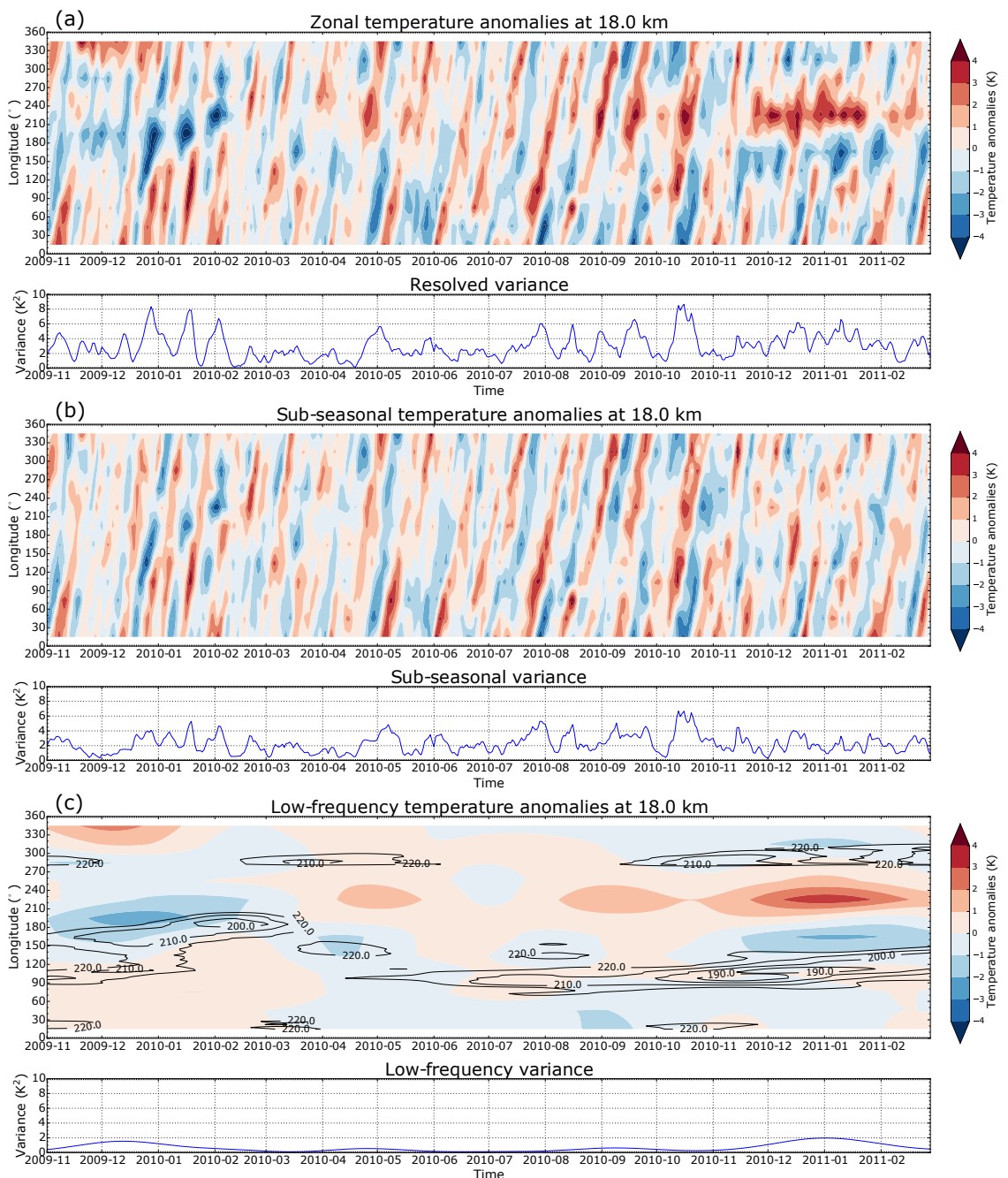

**Figure 3.** Hovmøller diagrams of zonal, sub-seasonal, and low-frequency temperature anomalies (top to bottom) at an altitude of 18 km from November 2009 to February 2011. Temporal evolutions of associated zonal mean variances are plotted on the bottom of each panel. Contour lines in panel (c) denote strong convection (low-frequency filtered OLR (W/m$^2$) averaged over $10°$S to $10°$N).

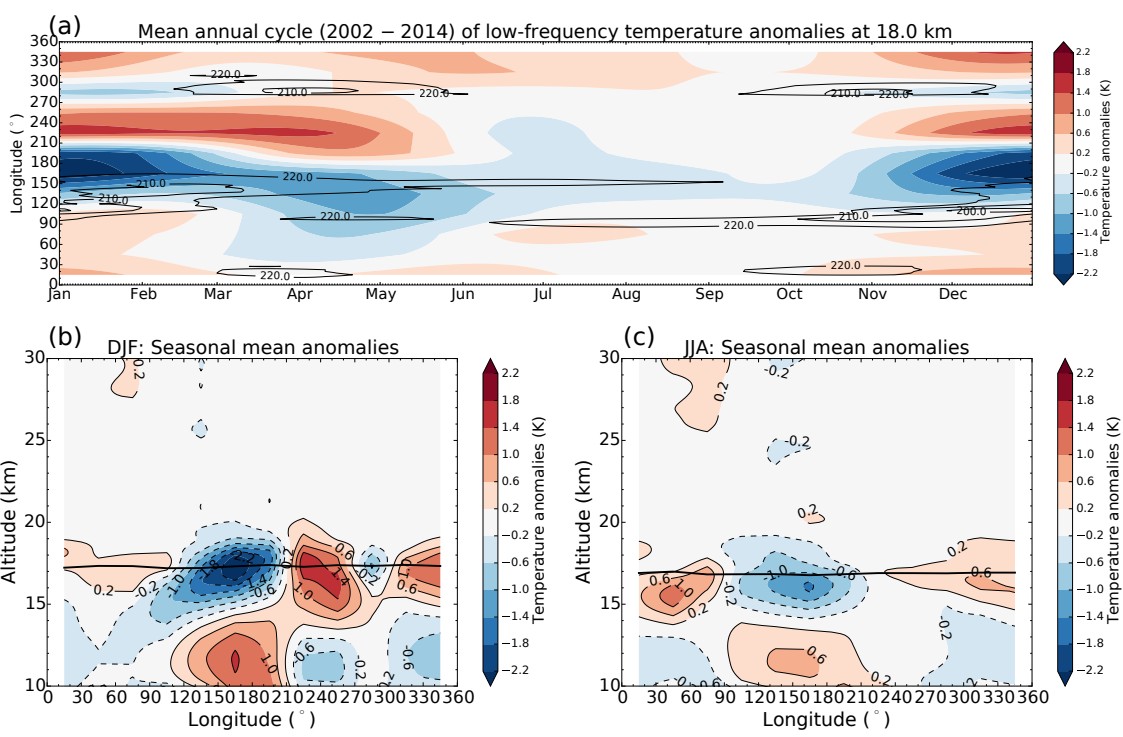

**Figure 4.** Hovmøller diagram of the mean annual cycle of low-frequency temperature anomalies at an altitude of 18 km (top panel) and associated seasonal mean anomalies as a function of longitude and altitude in DJF and JJA (bottom panels). Contour lines in panel (a) denote climatological strong convection (mean annual cycle of low-frequency filtered OLR, W/m$^2$). The black thick line in panels (b) and (c) indicates the lapse-rate tropopause.

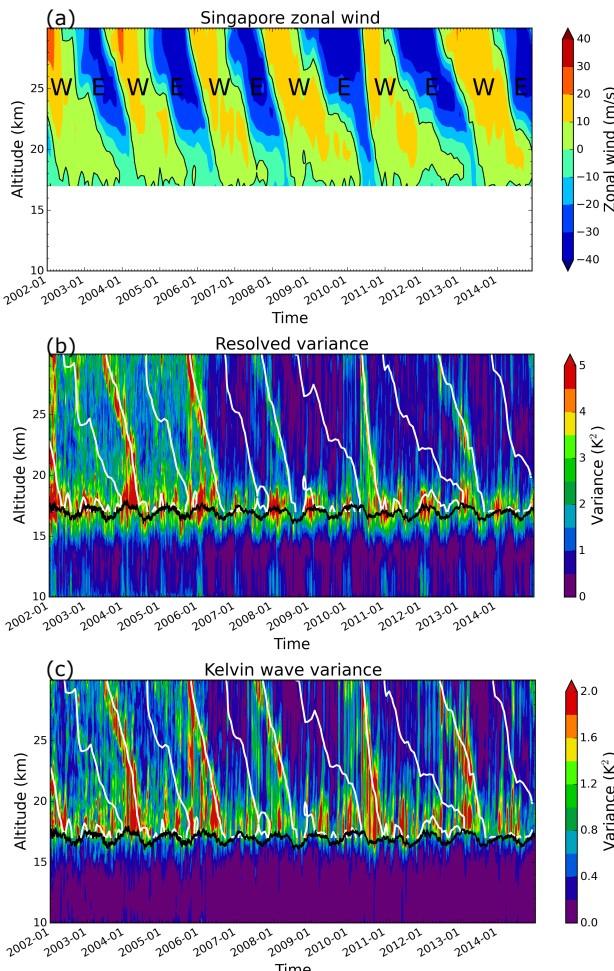

**Figure 5.** Time series of zonal mean wind speed above Singapore, smoothed resolved variance, and Kelvin wave variance as a function of altitude (top to bottom) from January 2002 to December 2014. E and W in the top panel refer to easterly and westerly wind. The black thick line in panels (b) and (c) indicates the lapse-rate tropopause, white thin contour lines indicate zero zonal wind speed above Singapore. Note the different color scales in panels (b) and (c).

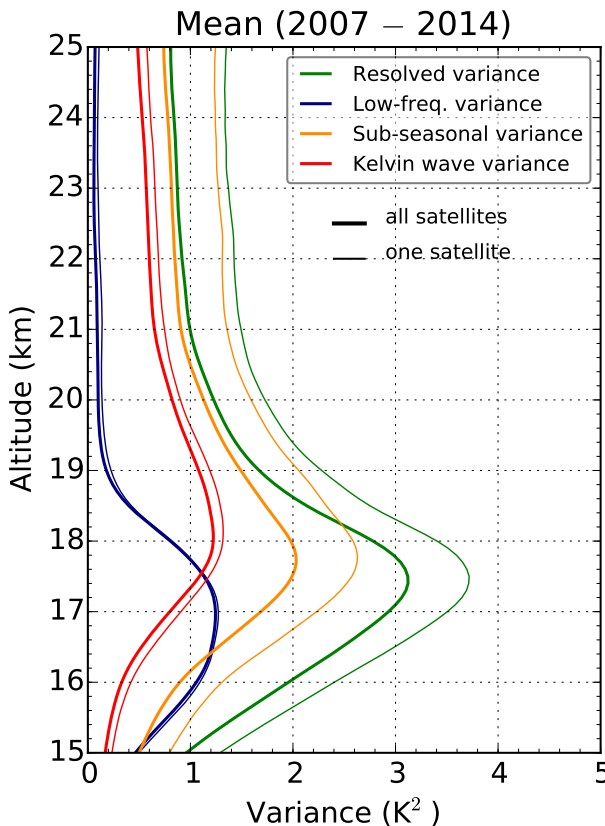

**Figure 6.** Mean of resolved variance (green), low-frequency variance (blue), sub-seasonal variance (orange), and Kelvin wave variance (red) as a function of altitude from 15 km to 25 km. Statistics of variances based on all satellite measurements is shown with thick lines, statistics of variances based on single satellite measurements is shown with thin lines. All statistics are obtained from January 2007 to December 2014.

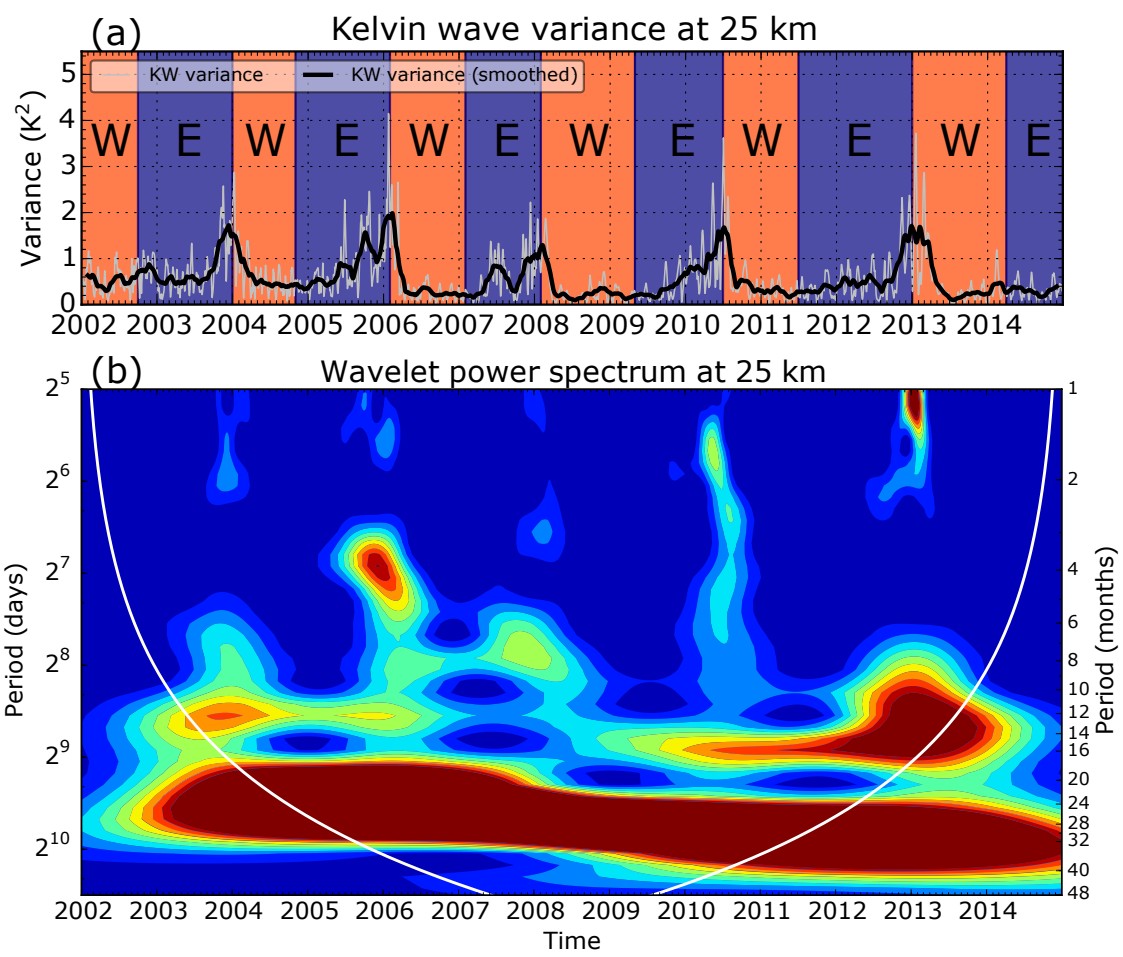

**Figure 7.** Time series of daily Kelvin wave variance (thin gray) and smoothed Kelvin wave variance (thick black) from January 2002 to December 2014 at an altitude of 25 km (top panel). Orange/blue background colors indicate westerlies (W)/easterlies (E) at 25 km above Singapore. Wavelet power spectrum of daily Kelvin wave variance at 25 km (bottom panel). The white line indicates the cone of influence. The period shown on the right y-axis (period in months) is calculated assuming 30 days per month.

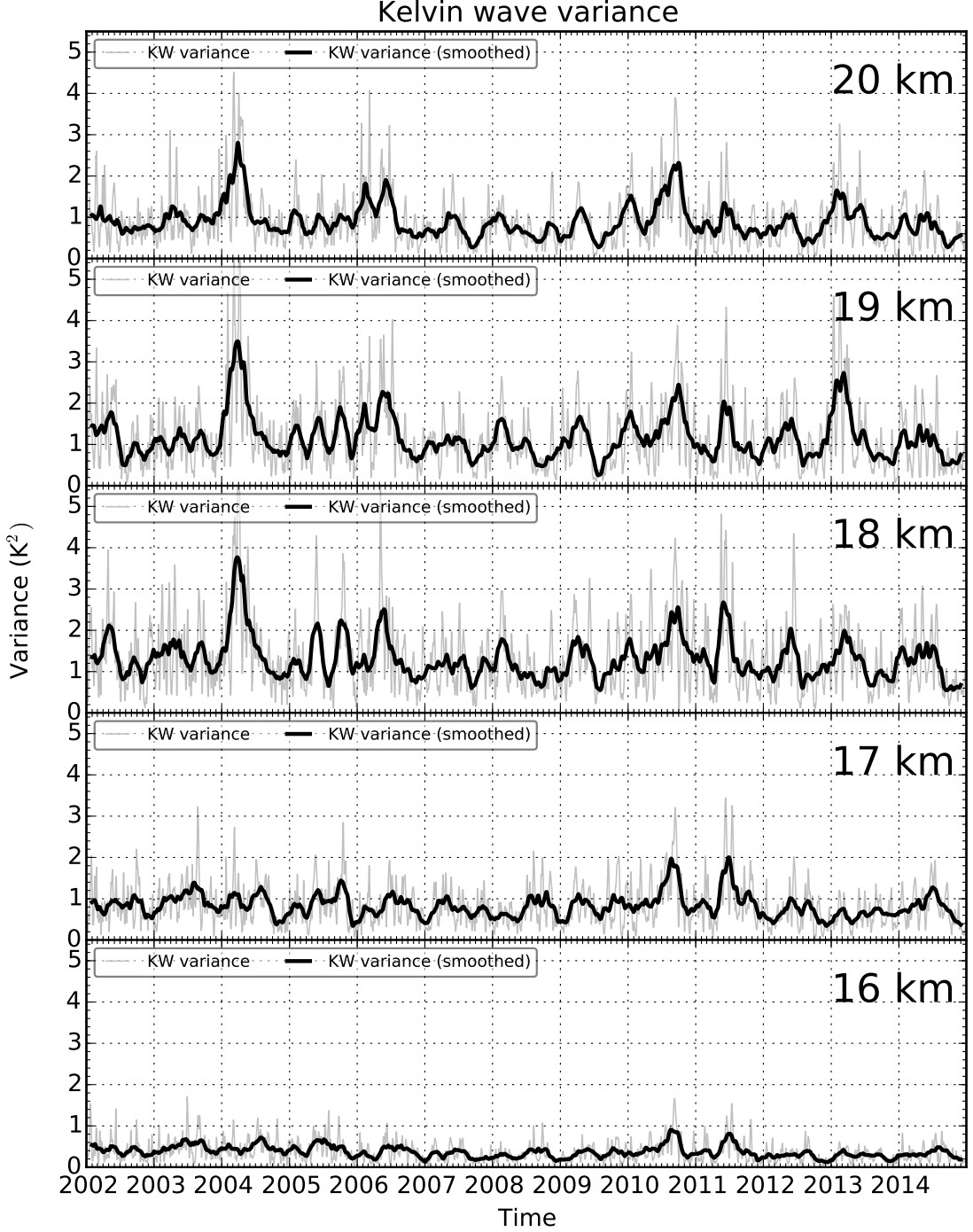

**Figure 8.** Time series of daily Kelvin wave variance (thin gray) and smoothed Kelvin wave variance (thick black) from January 2002 to December 2014 for every kilometer between 20 km (top) and 16 km (bottom).

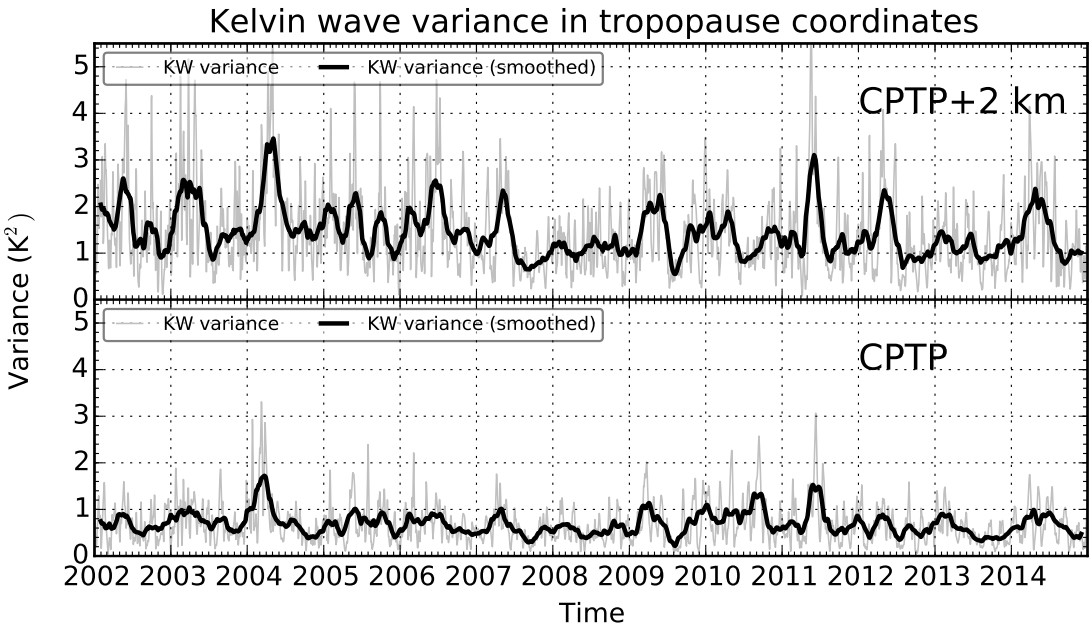

**Figure 9.** Time series of daily Kelvin wave variance (thin gray) and smoothed Kelvin wave variance (thick black) from January 2002 to December 2014 at the cold-point tropopause (bottom) and two kilometers above (top).

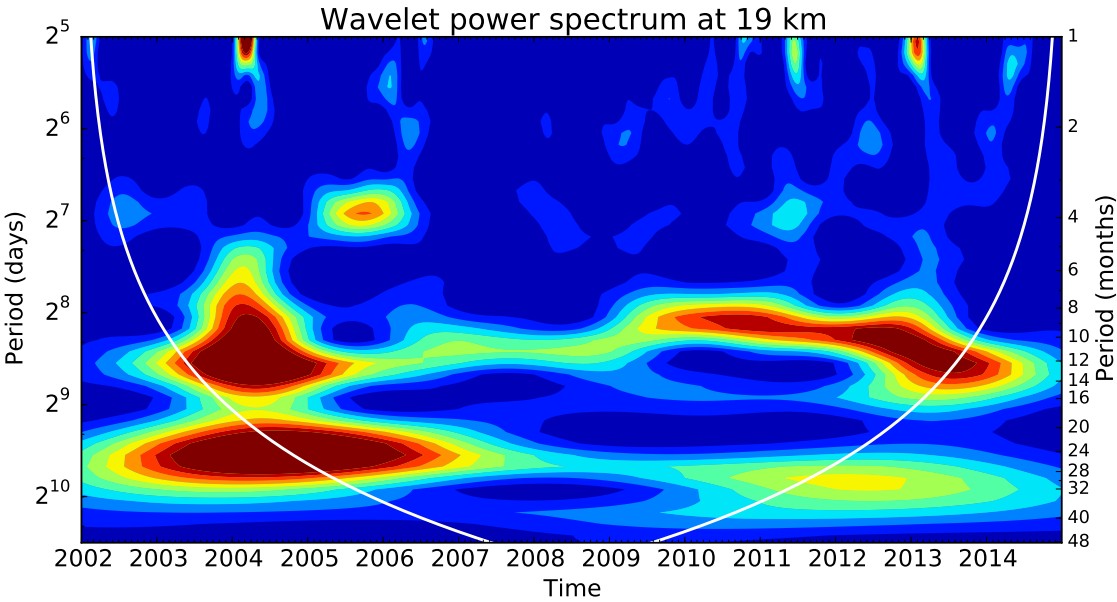

**Figure 10.** Wavelet power spectrum of daily Kelvin wave variance at 19 km from January 2002 to December 2014. The white line indicates the cone of influence. The period shown on the right y-axis (period in months) is calculated assuming 30 days per month.

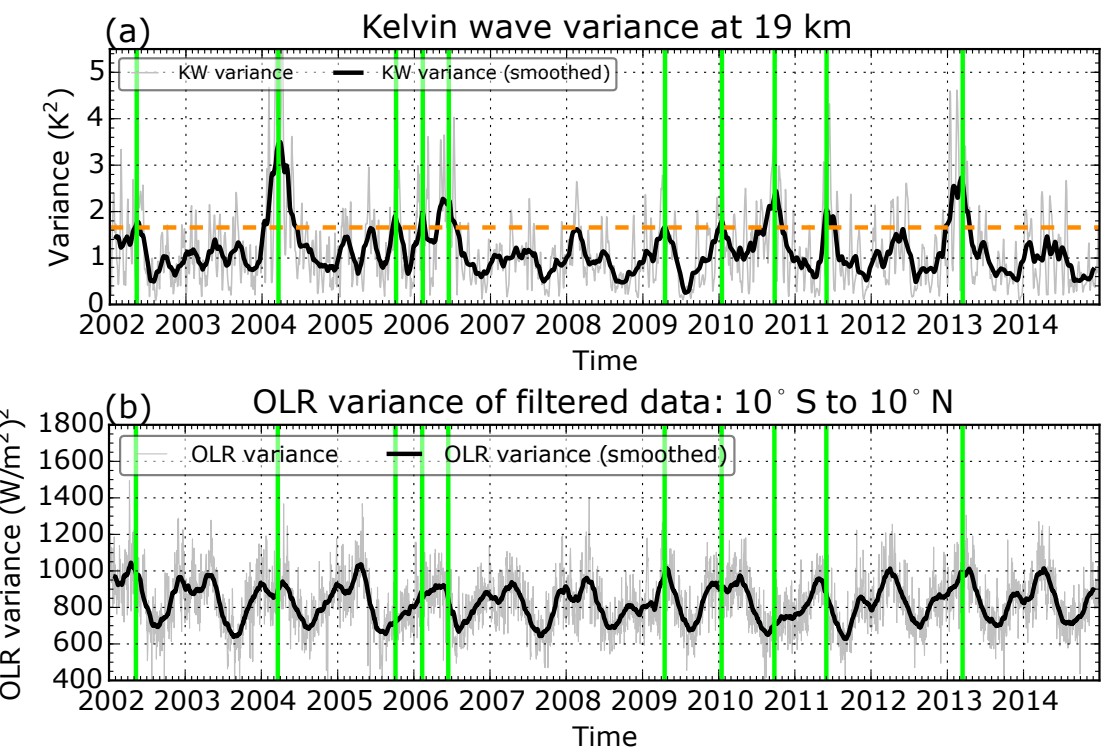

**Figure 11.** Time series of daily Kelvin wave variance (thin gray) and smoothed Kelvin wave variance (thick black) at 19 km (top panel) and time series of daily variances of high-pass filtered OLR data between 10°S and 10°N (bottom panel). Green lines indicate points of time with smoothed Kelvin wave variance outside of one standard deviation (1.66 K$^2$, indicated by the dashed yellow line).