# Peer review of "Tropical temperature variability and Kelvin wave activity in the UTLS from GPS RO measurements"

_Atmospheric Chemistry and Physics, 2016_

## Referee Comment (RC1) · Anonymous Referee #1 · 25 Aug 2016

Scherllin-Pirscher et al. use GPS radio occultation data to examine temperature variability in the tropical UTLS. Using spectral analysis, the authors identify contributions of temperature variability from quasi-stationary and sub-seasonal (<100 days) scales to the total variance. This study particularly focuses on Kelvin waves and their relationship with the QBO and convective forcing. The paper is nicely organized and easy to follow. It is interesting that their analysis on Kelvin wave activity in the tropopause does not show a QBO signal. I wonder if the result is affected by the calculation method to get Kelvin wave variance. Therefore, I recommend that the authors check the following points before publication in ACP.

My major concern is their definition of Kelvin waves. Although their definition of Kelvin

[Figure]

**[ACPD](ACPD)**

Interactive
comment

waves (zonal wavenumber 1-2 and frequency 7-30 days from the zonal space and time FFT) looks reasonable based on previous studies, I am not sure if the simple FFT filtering properly captures Kelvin wave variability quantitatively near the tropopause because amplitudes of tropopause waves strongly vary at different tropical regions. I agree that the inverse FFT of a filtered spectrum is a useful tool to construct characteristics of general zonal and vertical structure of Kelvin waves. This method, however, might not be a best way to quantify temperature variance associated with Kelvin waves, particularly for localized ones. If waves are nearly equally distributed over the tropics, the simple FFT should be sufficient to calculate time series of Kelvin wave variance. This will be the case for the stratosphere, but not for the tropopause layer.

I will use Figure 2 as an example. The authors explain that Kelvin wave variance in the tropopause is much smaller than the resolved total variance in Figure 2. To me, however, this figure demonstrates the simple 2D FFT Kelvin filter is not a proper way to calculate variance near the tropopause. For this event, it looks like Kelvin waves dominantly affect temperature anomalies above ∼15 km. The waves freely propagate over the whole tropics (non-localized) at ∼20-30 km, but they show localized distribution with a zonal wavenumber about 1.5 at 15-20 km. The localized amplitude in real space results in spread of a power spectrum, which will subsequently result in reduction of the amplitude after filtered inverse FFT. The exact same thing is shown in Figure 2b at 15-20 km. The smaller amplitude between 0-240E at 15-20 km in filtered anomalies (b) than in total anomalies (a) does not necessarily mean Kelvin wave variance is much smaller than the total variance; it would be just the localized signal cannot be captured in their method. Therefore, the whole conclusion about Kelvin wave variance near the tropopause over time is questionable. This includes the wavelet results since the wavelet method was applied after 2D Kelvin filtering.

My second concern is the use of a fixed altitude range for the tropopause layer. The tropopause is not a fixed level and varies with a seasonal cycle, QBO, etc. A fixed range includes mostly tropospheric characteristics when the tropopause is high. When

the tropopause is low, the fixed range more represents the stratosphere. Since temperature variability greatly changes from the troposphere to stratosphere, the variance calculation could be significantly affected by varying tropopause heights. It will be more reasonable to use a moving tropopause layer to calculate variance. For example, the authors can use relative heights to the monthly mean tropopause.

Minor comments: Term of "high-frequency": This term is not appropriate for disturbances shorter than 100 days. "Sub-seasonal" could be a replacement.

P4 L3: Add latitude resolution of gridded data

P4 L4: Add a statement like "This gridded data can only resolve waves with zonal wavenumbers up to 6." Also state the limitation of vertical scales that the dataset can resolve.

P4 L18: Gaussian filter filter → Gaussian filter

P4 L29: I assume only positive wavenumbers were included. To clarify this I suggest: "by selecting wavenumbers k=1 and 2 . . ." → "by selecting wavenumbers k=+1 and +2.." or explain only eastward signals were included. Alternatively a small box for Kelvin waves can be added in Figure 1.

P6 L21: ".. larger temperature variability before 2006 compared to after 2006." I think this implies that sub-grid scale variability (<1 day, < 60 degrees) is significant. Mention that.

Figure 1: The spectrum is too discrete between 10-50 days at zonal wavenumber 1. If my understanding is right, the whole time record was put to calculate the spectrum. Then, I think the spectrum will be more continuous than Figure 1. Why does the spectrum have separate peaks along the vertical axis?

"zonal anomalies" : The term is a bit confusing. I will rather use "anomalies," and describe the definition of anomalies. I presume anomalies are deviation from the time-mean zonal-mean.

Figure 10: Green, red, and blue lines are distracting. Those color lines seem subjective. If they are objectively decided, explain them. Otherwise I suggest deleting the color lines. Also Kelvin wave filters are for 7-30 days in (a), but OLR variance is for < 100 days in (b). Explain why they use different filter ranges.
* * *

---

## Referee Comment (RC2) · Anonymous Referee #2 · 26 Aug 2016

Summary

Scherllin-Pirscher et al. use radio occultation data collected between 2002 -2014 to analyse Kelvin wave activity and variability. The authors investigate temperature variance in the upper troposphere and lower stratosphere and examine the contributions to the total temperature variance from sources such as Kelvin waves and quasi-stationary wave activity. Scherllin-Pirscher et al. examine the relationship between peak Kelvin wave activity in the tropopause region and how it relates to deep convection. While the paper contains some interesting results and is well written, I would like the authors to consider the following points as they redraft their manuscript in preparation for publication in ACP.

[Figure]

Interactive
comment

General Comments

1) The authors use the phrase 'high frequency' throughout the manuscript which they define (Section 2.3) as those waves with periods shorter than 100 days. This is quite misleading given the usual definition of 'high frequency' waves refers to gravity waves, or perhaps, waves irresolvable in your data binning . Kelvin waves and gravity waves are known to drive the QBO, thus you should change this phrase to 'Kelvin-wave band' or 'monthly – seasonal band' or similar. And then your 'low frequency' band could be referred to as 'seasonal' or similar. Please change notation in Figures to reflect these changes in terminology too.

2) Usually upon using the Hayashi (1971) space-time spectral analysis method, authors retain Kelvin wave information in wavenumber-frequency space based on equivalent depths derived from Matsuno's (1966) shallow water equations. For example, see Wheeler & Kiladis (JAS 1999) for OLR and Ern et al. (ACP, 2008) for SABER and ECMWF temperatures in the stratosphere. Some comments on why a simple k=1,2 and 7<T<30 day periods filter is chosen in this manuscript is necessary. Also note that the wavenumber-frequency spectrum will change between the troposphere and stratosphere. Specifically, the spectrum will move to peak at higher frequencies in the stratosphere because higher frequency Kelvin waves propagate into the stratosphere more easily. Please investigate and discuss the wavenumber-frequency results at higher & lower altitude to confirm the validity of your filter limits and/or consider using equivalent depth filters instead.

3) Figure 2a: What role might the lapse-rate tropopause gradient play in the zonal anomalies shown here? That is, are you confident that you are removing all effects of the tropopause itself from this anomaly plot? I wonder in Figure 4b how or if the sharpness of the tropopause might influence these mean annual cycles of temperature anomalies – noting that the maximum positive and negative anomalies are right on the tropopause altitude? Do these maximum anomalies change in altitude following the seasonal cycle of tropopause altitude itself?

Minor Comments

P7 line 20: What are you defining as 'high frequency' here?

Figure 3: Worth noting that W=westerly, E=easterly to avoid confusion between westerly/westward.

Figure 10: Replot the boxes on top of the green, red, blue lines as it's not well presented at the moment.

Technical corrections, grammar, etc.

P2, Line 10: 'were theorised by Matsuno'

P2, Line 16: 'important role in the stratosphere –'

P3, line 13: 'Due to the RO measurement'

P3, line 16 & 17: change 'are' to 'were'

P3, line 24: 'information on convection'

P3, line 27: 'vertically-resolved'

P4 line 18 'Gaussian filter (with'

P8 line 4 & 5: Boreal or austral spring 2003 & 2004?

P9 line 6 'these data ideal for characterizing'

---

## Author Comment (AC1) · 26 Oct 2016

We thank the reviewer for the review and his/her helpful and constructive comments which we fully took into account in the revision of the paper. Please see our detailed response below (the original remarks of the referee are in italics).

**Major comments**

1. *My major concern is their definition of Kelvin waves. Although their definition of Kelvin waves (zonal wavenumber 1–2 and frequency 7–30 days from the zonal*

*space and time FFT) looks reasonable based on previous studies, I am not sure if the simple FFT filtering properly captures Kelvin wave variability quantitatively near the tropopause because amplitudes of tropopause waves strongly vary at different tropical regions. I agree that the inverse FFT of a filtered spectrum is a useful tool to construct characteristics of general zonal and vertical structure of Kelvin waves. This method, however, might not be a best way to quantify temperature variance associated with Kelvin waves, particularly for localized ones. If waves are nearly equally distributed over the tropics, the simple FFT should be sufficient to calculate time series of Kelvin wave variance.*

We thank the referee for highlighting this important issue. We reassessed the definition of atmospheric Kelvin waves and decided to filter Kelvin waves with wavenumbers $k = +1$ to +6 and periods from $4$ to 30 days. These filter widths are determined by our sampling strategy as well as by the spatial resolution of our daily-mean fields. Furthermore, we will follow Kim and Son (2012) and filter Kelvin waves between 8 m and 240 m equivalent depths.

Our new method is now able to also capture local enhancements of Kelvin wave power in the tropical tropopause layer. We will replot Fig. 1 (where we will include equivalent depths of 8 m and 240 m) as well as Fig. 2b, Fig. 5c, and Fig. 6 to Fig. 10.

In the manuscript, we will add the following sentence in Sect. 2.3:

We isolate Kelvin wave activity by selecting wavenumbers $k = +1$ to $+6$, periods from 4 to 30 days, and equivalent depths from 8 m to 240 m.

2. *My second concern is the use of a fixed altitude range for the tropopause layer. The tropopause is not a fixed level and varies with a seasonal cycle, QBO, etc. A fixed range includes mostly tropospheric characteristics when the tropopause is high. When the tropopause is low, the fixed range more represents the strato-sphere. Since temperature variability greatly changes from the troposphere to*

*stratosphere, the variance calculation could be significantly affected by varying tropopause heights. It will be more reasonable to use a moving tropopause layer to calculate variance.*

In order to investigate the sensitivity of Kelvin wave activity in the tropical tropopause region on different coordinate systems, we calculated and compared Kelvin wave activity in fixed-altitude and in cold-point tropopause coordinates. We find that temporal variability of Kelvin wave variance at the cold-point tropopause is in very good agreement with variability at 17 km. Furthermore, Kelvin wave variance 2 km above the cold-point tropopause is very similar to Kelvin wave variance at 19 km. We will include both these plots as new Fig. 9 in the manuscript and add a discussion in Sect. 3.3:

To assess whether this temporal variability should be attributed to temporal variations of the tropopause rather than to Kelvin wave activity itself, we calculated Kelvin wave variance in cold-point tropopause coordinates. Figure 9 shows results at the cold-point tropopause and two kilometers above. Comparison to Fig. 8 shows very similar temporal evolutions. The time series at the tropopause and 2 km above (Fig. 9) are virtually similar to those at 17 km and 19 km in altitude coordinate (Fig. 8), respectively. Again, no clear periodicity of Kelvin wave activity can be found.

**Minor comments**

1. *Term of "high-frequency": This term is not appropriate for disturbances shorter than 100 days. "Sub-seasonal" could be a replacement.*

   We agree with the reviewer. We will replace "high-frequency variability" by "sub-seasonal variability" in the entire manuscript (text and figures).

2. *P4 L3: Add latitude resolution of gridded data*

   We averaged over all high-quality profiles within 10°S and 10°N and performed our analysis for this tropical band (with 20° width). Therefore, our gridded fields are not latitudinally resolved. To make this more clear we will rewrite this sentence. It will read:

   Tropical (10°S to 10°N) temperature profiles from GPS RO are gridded in longitude, altitude, and time (no further latitudinal gridding).

3. *P4 L4: Add a statement like "This gridded data can only resolve waves with zonal wavenumbers up to 6." Also state the limitation of vertical scales that the dataset can resolve.*

   Thank you for this suggestion. We will include this statement in the manuscript.

   The vertical resolution of GPS RO temperature profiles ranges from about 100 m in the troposphere to about 1.5 km in the stratosphere. This is mentioned in Section 2.1 in the manuscript and implies that individual levels of our analysis are not independent from each other. We will add

   Since underlying GPS RO profiles have a vertical resolution of 0.1–1.5 km (see Sect. 2.1), adjacent vertical levels are not fully independent of each other.

4. *P4 L18: Gaussian filter filter → Gaussian filter*

   done

5. *P4 L29: I assume only positive wavenumbers were included. To clarify this I suggest: "by selecting wavenumbers k=1 and 2 . . . " → "by selecting wavenumbers k=+1 and +2. . . " or explain only eastward signals were included. Alternatively a small box for Kelvin waves can be added in Figure 1.*

   Good point. We will add the plus sign in the text (the entire sentence is given in reply to major comment 1).

6. *P6 L21: "... larger temperature variability before 2006 compared to after 2006." I think this implies that sub-grid scale variability (<1 day, <60 degrees) is significant. Mention that.*

   We agree with the reviewer that sub-grid scale variability is significant. We already mentioned that in the manuscript, P6, L26/27, which reads:

   This can be explained by noting that much of sub-seasonal variance is related to sub-grid-scale (smaller than $30°$) variation (as shown in Randel and Wu, 2005), which is under-sampled by one satellite.

7. *Figure 1: The spectrum is too discrete between 10–50 days at zonal wavenumber 1. If my understanding is right, the whole time record was put to calculate the spectrum. Then, I think the spectrum will be more continuous than Figure 1. Why does the spectrum have separate peaks along the vertical axis?*

   It is correct that the spectrum is based on the entire RO record. Please note, however, the very small contour line interval (only $0.0002$ K$^2$). In some cases, this small, linear contour interval can produce separate peaks in the spectrum.

   Figure 1 shows that most power is concentrated at low frequencies (annual cycle and inter-annual variability). To better show the general picture of spectral power, we will replot this figures using logarithmic contour interval using contour lines of $(0.0001, 0.00025, 0.0005, 0.001, \ldots, 10, 25, 50)$ K$^2$.

8. *"zonal anomalies": The term is a bit confusing. I will rather use "anomalies", and describe the definition of anomalies. I presume anomalies are deviation from the time-mean zonal-mean.*

   In Fig. 2 and 3, zonal temperature anomalies are obtained from subtracting the daily-mean zonal-mean profile. To clarify how these zonal anomalies were obtained, we will write in the text:

[Figure]

Strongest amplitudes are observed near the tropopause, where the magnitude of zonal anomalies (i.e., anomalies relative to the daily-mean zonal-mean) exceeds 5 K and that of Kelvin waves exceeds 2 K.

Figure 3 shows temporal evolution of zonal temperature anomalies (relative to daily-mean zonal-mean) from November 2009 to February 2011 at 18 km.

Figure 4b and 4c show low-frequency temperature anomalies averaged over DJF and JJA, respectively. We will remove "zonal" in these plot titles and rather write "DJF: Seasonal mean anomalies" and "JJA: Seasonal mean anomalies".

9. *Figure 10: Green, red, and blue lines are distracting. Those color lines seem subjective. If they are objectively decided, explain them. Otherwise I suggest deleting the color lines. Also Kelvin wave filters are for 7–30 days in (a), but OLR variance is for <100 days in (b). Explain why they use different filter ranges.*

Yes, different colors were chosen subjectively. We will remove red and blue lines but indicate peaks in Kelvin wave variance outside of one standard deviation with green lines.

We tested the hypothesis whether sub-seasonal fluctuations in convection (including different types of waves) could explain irregularly distributed peaks in Kelvin wave variance. We did not use OLR variance using the very same filter width because Kelvin waves at and above the tropical tropopause are not convectively coupled waves anymore. Kelvin waves in OLR data are of smaller scale and faster periods compared to our Kelvin waves close to the tropical tropopause (Wheeler and Kiladis 1999). Furthermore we found that these waves contribute very little variance and are also included in these high-pass filtered waves.

**References**

J. Kim and S.-W. Son. Tropical cold-point tropopause: Climatology, seasonal cycle, and intraseasonal variability derived from COSMIC GPS radio occultation measurements. *J. Climate*, 25(15):5343–5360, doi:10.1175/JCLI-D-11-00554.1, 2012.

M. Wheeler and G. N. Kiladis. Convectively coupled equatorial waves: Analysis of clouds and temperature in the wavenumber–frequency domain. *J. Atmos. Sci.*, 56: 374–399, 1999.

---

## Author Comment (AC2) · 26 Oct 2016

We thank the reviewer for the review and his/her helpful and constructive comments which we fully took into account in the revision of the paper. Please see our detailed response below (the original remarks of the referee are in italics).

**General comments**

1. *The authors use the phrase "high frequency" throughout the manuscript which they define (Section 2.3) as those waves with periods shorter than 100 days. This*

[Figure]

*is quite misleading given the usual definition of "high frequency" waves refers to gravity waves, or perhaps, waves irresolvable in your data binning . Kelvin waves and gravity waves are known to drive the QBO, thus you should change this phrase to "Kelvin-wave band" or "monthly – seasonal band" or similar. And then your "low frequency" band could be referred to as "seasonal" or similar. Please change notation in Figures to reflect these changes in terminology too.*

Thanks for pointing at this miswording! We agree with the reviewer and will replace "high-frequency" variability by "sub-seasonal" variability in the entire manuscript (text and figures). Since "low-frequency" variability does not only include to "seasonal" variability but also inter-annual variability, we will not change this notation.

2. *Usually upon using the Hayashi (1971) space-time spectral analysis method, authors retain Kelvin wave information in wavenumber-frequency space based on equivalent depths derived from Matsuno's (1966) shallow water equations. For example, see Wheeler & Kiladis (JAS 1999) for OLR and Ern et al. (ACP, 2008) for SABER and ECMWF temperatures in the stratosphere. Some comments on why a simple $k = 1$,2 and $7 < T < 30$ day periods filter is chosen in this manuscript is necessary. Also note that the wavenumber-frequency spectrum will change between the troposphere and stratosphere. Specifically, the spectrum will move to peak at higher frequencies in the stratosphere because higher frequency Kelvin waves propagate into the stratosphere more easily. Please investigate and discuss the wavenumber-frequency results at higher & lower altitude to confirm the validity of your filter limits and/or consider using equivalent depth filters instead.*

We thank the referee for highlighting this important issue. We reassessed the definition of atmospheric Kelvin waves and decided to filter Kelvin waves with wavenumbers $k = +1$ to $+6$ and periods from 4 to 30 days. These filter widths are determined by our sampling strategy as well as by the spatial resolution of

our daily-mean fields. Furthermore, we will follow Kim and Son (2012) and filter equivalent depths from 8 m to 240 m. Our new filter will be able to also capture local enhancements of Kelvin wave power, which is of particular importance in the tropical tropopause layer. We will replot Fig. 1 (where we will include maximum and minimum equivalent depths of 8 m and 240 m, respectively, and use logarithmic contour intervals instead of linear ones) as well as Fig. 2b, Fig. 5c, and Fig. 6 to Fig. 10.

In the manuscript, we will add the following sentence in Sect. 2.3:

We isolate Kelvin wave activity by selecting wavenumbers $k = +1$ to +6, periods from 4 to 30 days, and equivalent depths from 8 m to 240 m.

3. *Figure 2a: What role might the lapse-rate tropopause gradient play in the zonal anomalies shown here? That is, are you confident that you are removing all effects of the tropopause itself from this anomaly plot? I wonder in Figure 4b how or if the sharpness of the tropopause might influence these mean annual cycles of temperature anomalies – noting that the maximum positive and negative anomalies are right on the tropopause altitude? Do these maximum anomalies change in altitude following the seasonal cycle of tropopause altitude itself?*

We agree with the reviewer. There could be some effect of temperature variability due to the seasonal cycle of tropopause characteristics. Figs. 4b,c, which show the vertical structure of seasonal temperature anomalies, reveal, however, that the amplitude of DJF anomalies is notably larger than that in JJA. Furthermore, the seasonality shown in Fig. 4b is still distinctive even if we follow tropopause-relative coordinates.

We also tested the possible influence of the change in tropopause altitude on Kelvin wave activity by comparing temperature anomalies in altitude coordinates and tropopause-relative coordinates. We find that temporal variability of Kelvin wave variance at the cold-point tropopause is in very good agreement with variability at 17 km. Furthermore, Kelvin wave variance 2 km above the cold-point tropopause is very similar to Kelvin wave variance at 19 km. Therefore, the change of tropopause altitude seems to have minor effect. We will include both timeseries of Kelvin wave activity at cold-point tropopause and 2 km above as new Fig. 9 in the manuscript and will add a discussion in Sect. 3.3:

To assess whether this temporal variability should be attributed to temporal variations of the tropopause rather than to Kelvin wave activity itself, we calculated Kelvin wave variance in cold-point tropopause coordinates. Figure 9 shows results at the cold-point tropopause and two kilometers above. Comparison to Fig. 8 shows very similar temporal evolutions. The time series at the tropopause and 2 km above (Fig. 9) are virtually similar to those at 17 km and 19 km in altitude coordinate (Fig. 8), respectively. Again, no clear periodicity of Kelvin wave activity can be found.

**Minor comments**

1. *P7 line 20: What are you defining as "high frequency" here?* .

   "High-frequency" amplitude variations were meant to be temporal variations in the waves' amplitude.

   We will rewrite this sentence to make it more clear. It will read:

   Kelvin wave variance in Fig. 7a shows temporal variations in the waves' amplitude that are strongly modulated by the QBO, with enhanced wave activity in periods of transition from easterly to westerly stratospheric wind (westerly shear zones).

2. *Figure 3: Worth noting that W=westerly, E=easterly to avoid confusion between westerly/westward.*

We assume that the referee refers to Figure 5, where we will add in the figure caption:

E and W in the top panel refer to easterly and westerly wind.

3. *Figure 10: Replot the boxes on top of the green, red, blue lines as it's not well presented at the moment.*

   Since different colors were chosen subjectively, we will remove red and blue lines in this figure (remark of reviewer 1). In order to indicate peaks in Kelvin wave variance outside of one standard deviation, we will only plot green lines. We will plot boxes on top of these green lines.

**Technical corrections, grammar, etc.**

1. *P2, Line 10: "were theorised by Matsuno"*—done

2. *P2, Line 16: "important role in the stratosphere – "*—done

3. *P3, line 13: "Due to the RO measurement"*—done

4. *P3, line 16 & 17: change "are" to "were"*—done

5. *P3, line 24: "information on convection"*—done

6. *P3, line 27: "vertically-resolved"*—done

7. *P4, line 18 "Gaussian filter (with. . . "*—done

8. *P8, line 4 & 5: Boreal or austral spring 2003 & 2004?*—in boreal spring. We added "boreal" in these sentences.

9. *P9 line 6 "these data ideal for characterizing"*—done
**References**

J. Kim and S.-W. Son. Tropical cold-point tropopause: Climatology, seasonal cycle, and intraseasonal variability derived from COSMIC GPS radio occultation measurements. *J. Climate*, 25(15):5343–5360, doi:10.1175/JCLI-D-11-00554.1, 2012.